# Mapping knowledge gaps in marine diversity reveals a latitudinal gradient of missing species richness

André Menegotto [1] & Thiago F. Rangel [2]

A reliable description of any spatial pattern in species richness requires accurate knowledge about species geographical distribution. However, sampling bias may generate artefactual absences within species range and compromise our capacity to describe biodiversity patterns. Here, we analysed the spatial distribution of 35,000 marine species (varying from copepods to sharks) to identify missing occurrences (gaps) across their latitudinal range. We find a latitudinal gradient of species absence peaking near the equator, a pattern observed in both shallow and deep waters. The tropical gap in species distribution seems a consequence of reduced sampling effort at low latitudes. Overall, our results suggest that spatial gaps in species distribution are the main cause of the bimodal pattern of marine diversity. Therefore, only increasing sampling effort at low latitudes will reveal if the absence of species in the tropics, and the consequent dip in species richness, are artefacts of sampling bias or a natural phenomenon.

[1] Programa de Pós-Graduação em Ecologia e Evolução, Universidade Federal de Goiás, Goiânia 74690-900 GO, Brazil. [2] Departamento de Ecologia, Universidade Federal de Goiás, Goiânia 74690-900 GO, Brazil. Correspondence and requests for materials should be addressed to A.M. (email: andre.menegotto@gmail.com)

arge-scale spatial patterns of species richness, such as the latitudinal gradient of biodiversity, are an emergent property of the overlap of the geographic distribution of individual species[1]. Therefore, a reliable description of the spatial patterns in species richness requires accurate knowledge of the range of all species. However, the current knowledge of the geographic distribution of most species is largely incomplete (the Wallacean shortfall[2,3]), especially in regions of difficult and costly accessibility[4]. Because knowledge gaps of species distribution are not randomly distributed across space, but cluster in undersampled regions, the spatial patterns in species richness are statistically biased. Although ecologists have developed analytical techniques to estimate total species richness in a community or region (for instance using incidence of rare species[5]), these techniques usually do not reveal the identity of the potentially missing species, nor do they consider the distribution of missing species in large-scale patterns of species richness (but see occupancy models[6,7]).

Determining whether the lack of occurrence records of a species in a given locality is a true absence or a sampling artefact is one of the main challenges faced by geographical ecologists[8]. However, if the distribution of species can be assumed to be contiguous across space at a given scale (see Supplementary Note 1; Supplementary Fig. 1), the spatial gaps in recorded presences provide an indication of the magnitude of sampling bias in a given region. For example, if a particular species is recorded at latitudes 0°, 10°, 30° and 40°, the lack of occurrence records at latitude 20° suggests an artefactual absence, especially if the sampling effort at that latitude is known to be insufficient.

By identifying the spatial gaps in the distribution of species across latitudes it is possible to map the latitudinal distribution of potential Wallacean shortfall (Fig. 1), and to explore how and where the missing records are affecting the species richness estimate. Conversely, the absence of a species in an area of high sampling effort and inventory completeness suggests a true gap in species distribution, which is a valuable observation to study ecological processes[9]. Even so, few studies have explored the spatial distribution of missing species occurrence[8,10] and, to our best knowledge, no study has yet evaluated how the patterns of species absence may affect the richness estimate at global-scale.

Here we analysed an extensive data set of more than 3 million occurrence records of 35,000 marine species, to evaluate the spatial pattern in the distributional gaps across their latitudinal range. Recent studies have proposed that the latitudinal gradient of species richness in the marine environment is bimodal[11,12], with a dip in species richness near the equator, in marked contrast to the pattern observed in the continental realm. Here we evaluated if this dip is caused by a natural decrease in the number of species, or by an artefact of missing records of existing species. We estimated the richness of missing species by associating the observed discontinuities in species geographic ranges and the distribution of sampling effort at 5° latitudinal bands. We concluded that a bimodal pattern in marine species richness arise because of a tropical gap in species distribution, which is strongly associated with low sampling effort in the tropics.

## Results

### Species richness, spatial gaps and sampling efficiency.
According to available records of occurrences of marine species, the latitudinal gradient in marine species richness is bimodal, with peaks at mid-latitudes (total number of species between −25° and −20°: 7129; between 20° and 25°: 6320) and a dip at the equator (Fig. 2a). Although not all groups have a bimodal richness distribution (copepods, stony corals and tuna fishes, have a unimodal distribution), this general pattern of species richness

was observed for most taxa that we analysed (Supplementary Fig. 2).

The tropical dip in species richness coincides with a spatial gap (disjunction) in the latitudinal distribution of 11,432 species (32.80% of all species analysed). If not artefactual, such result would indicate a surprisingly common pattern of species occurring in both hemispheres, while not occurring in the tropical belt (Fig. 2a). In fact, there is a total of 5088 species absent from the equator (more precisely between 0° and 5°), which is 1.78 times the number of species recorded at the same latitude (2866 species). This unimodal pattern of tropical absence peaking at the equator can be observed in all taxonomic groups (Supplementary Fig. 2).

The average number of sampling events (unique combination of latitude, longitude and sampling date) was highest at mid-latitudes, especially in the northern hemisphere (Fig. 2b). In the southern mid-latitudes (between −35° and −30°), there was an average (± standard deviation) of 9463 ± 15,626 sampling events, whereas in the northern mid-latitudes (between 50° and 55°) the mean number of sampling events reached 21,644 ± 16,455, with a maximum of 55,346 sampling events for copepods. In fact, the number of sampling events in northern mid-latitudes was substantially higher than in any other region, for all taxonomic groups (Supplementary Fig. 3). Conversely, the mean number of sampling events at the equator was not even higher than 700 (between −5° and 0°: 592 ± 574; between 0° and 5°: 628 ± 835), with a maximum of 2688 samplings events for tuna fishes.

To evaluate if the spatial pattern of species absence is a byproduct of the unequal sampling effort we applied two different estimates of inventory completeness for each latitudinal band (sample coverage and species accumulation curve; see Methods section). As expected, both estimates of inventory completeness revealed an unequal sampling efficiency across the globe. On average, the level of completeness was generally high at mid-latitudes, especially in those with the highest sampling effort (Fig. 2b). In the northern mid-latitudes, for example, the estimates of inventory completeness reached values near 1.0 for almost all groups analysed. Conversely, most taxa remain vastly undersampled at the equator. Considering both completeness estimation methods (sample coverage and species accumulation curve), only vertebrate fishes have an inventory completeness relatively high at the equator (above 0.9; Supplementary Figs. 4–5). However, for most taxonomic groups the level of completeness was lower than 0.8 (0.74 ± 0.27) as estimated by sample coverage, and lower than 0.5 (0.45 ± 0.33) as estimated by species accumulation curve (Fig. 2b). The inventory completeness of phylum Porifera, for instance, was estimated at 0.22 by sample coverage, and approximately zero by species accumulation curve (Supplementary Figs. 4–5). We also tested how the spatial resolution of analysis affects the estimated latitudinal pattern in inventory completeness. Although completeness estimates tend to increase at lower resolutions, the tropics are found to be incompletely sampled at all spatial resolutions (Supplementary Fig. 6).

### Latitudinal pattern across bathymetric strata.
Because the latitudinal gradient in species richness, as well as sampling intensity, may vary across depth[12,13], we evaluated if the general pattern described above is consistent across three different bathymetric strata: euphotic (0–200 m), bathyal (200–2000 m) and abyssal (2000–6500 m). We found that latitudinal patterns in species richness, spatial gaps, and sampling efficiency in the euphotic and bathyal strata were similar to that described for the entire ocean (Fig. 3a–d), as 89.23% of all records analysed came from shallow waters (Supplementary Note 2). The average sampling effort in

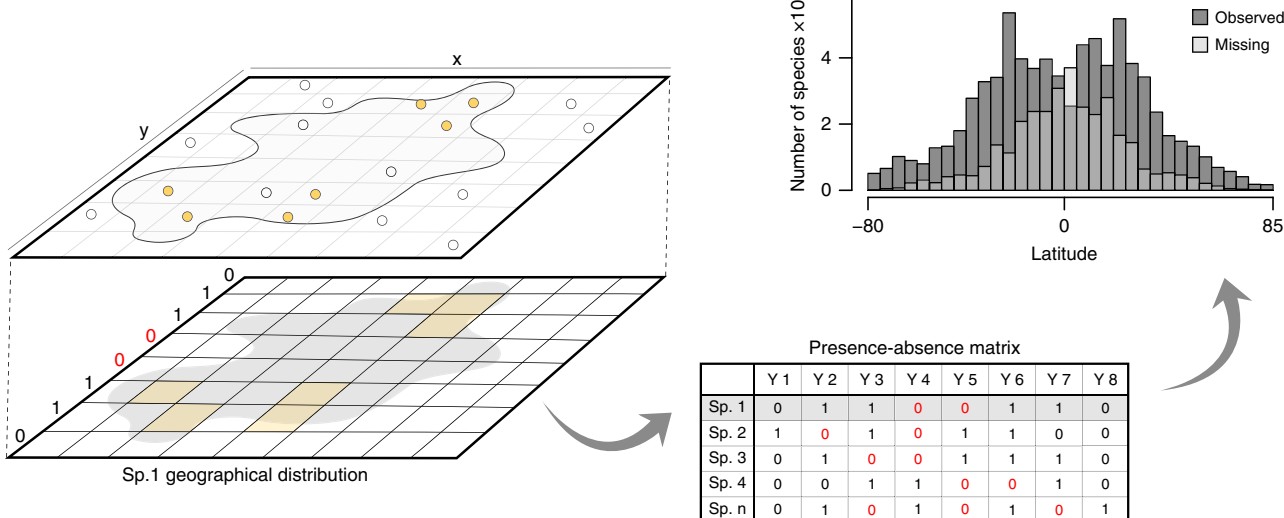

**Fig. 1** From species sampling to spatial richness pattern. The lack of occurrence records of a species in a location (white dots) may be caused by its true absence (geographic distribution) or by poor sampling (artefactual absence). In this example, the irregular polygon in the extruded map represents the true distribution of a species. Researchers only have access to occurrence records (yellow dots), which are then mapped in discrete grid cells (yellow cells, lower map), which may contain discontinuities in the estimated species range if the inventory is not sufficient. The observed presence (1) and absence (0) of species in latitudinal bands are recorded along the left margin of the lower map, in which red zeros indicate the spatial gaps in the latitudinal distribution of the species. By compiling occurrence data for multiple species, we can quantify not only the richness pattern but also the frequency of spatial gaps (note that colours are mixed where histograms overlap; histogram based on real Ophiuroidea data)

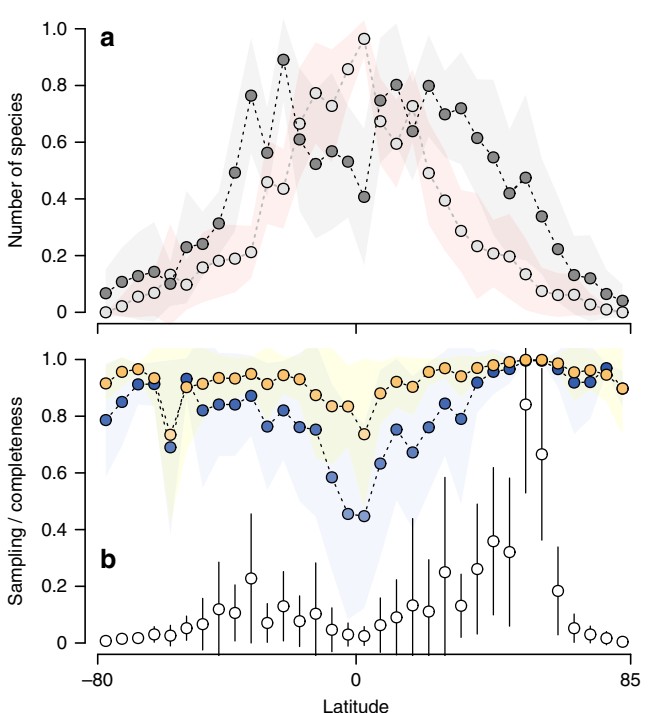

**Fig. 2** Patterns of latitudinal variation in species richness, spatial gaps and sampling efficiency for marine taxa. Circles represent mean values based on results of 10 taxonomic groups. Shaded area and vertical bars represent standard deviation (s.d.). Data were standardised by the maximum observed value to range between 0 and 1. **a** Mean (± s.d.) latitudinal variation in number of observed species (dark grey circles) and number of missing species (light grey circles). Notice that the number of missing species exceeds the number of recorded species at lower latitudes. **b** Mean (± s.d.) latitudinal variation in sampling effort (number of unique sampling events; open circles) and two estimates of inventory completeness (sample coverage: orange circles; species accumulation curve: dark blue circles)

the deep sea is also highest at mid-latitudes of the northern hemisphere and comparatively lower in the tropics (Fig. 3f). Thus, the distribution of spatial gaps also peaked within the tropics (maximum of 361 absences for stony corals between −20° and −15°; Fig. 3e). However, at greater depths the spatial distribution of observed species richness was highly heterogeneous among the groups (Supplementary Figs. 7–16), with no clear latitudinal pattern (Fig. 3e).

## Discussion

Contrary to the paradigm of a unimodal richness pattern observed in the continental realm, recent studies have proposed that the latitudinal gradient of species diversity in the marine environment is bimodal, with a dip in species richness near the equator[11,12]. Indeed, by analysing an extensive data set of more than 3 million presence records, for 10 different taxonomic groups, we found a bimodal pattern in species richness. However, a careful analysis of gaps in the latitudinal distribution of species revealed that the dip in species richness at lower latitudes is mostly caused by missing species records. In fact, there is an increase of gaps in species distribution toward the equator, among all 10 taxonomic groups analysed, regardless of taxonomic level or body size. Although many studies have previously examined the latitudinal gradient of diversity in the marine realm[11,12,14–16], our study is the first to explore the contiguity of species latitudinal distribution and to reveal a latitudinal gradient of species absences.

The absence of many marine species at the equator might suggest, at first glance, a general pattern of antitropical distribution, a phenomenon in which identical or closely related species occupy both hemispheres but are absent from the tropics[17]. This phenomenon is not rare in the marine system, and has been described for many taxonomic groups since the expeditions of Captain J. C. Ross in the 19th century[18]. It is noteworthy, however, that antitropical distribution commonly involves taxonomic groups above species level (i.e., genera or family). Although there are known marine species with antitropical distribution[19,20], there

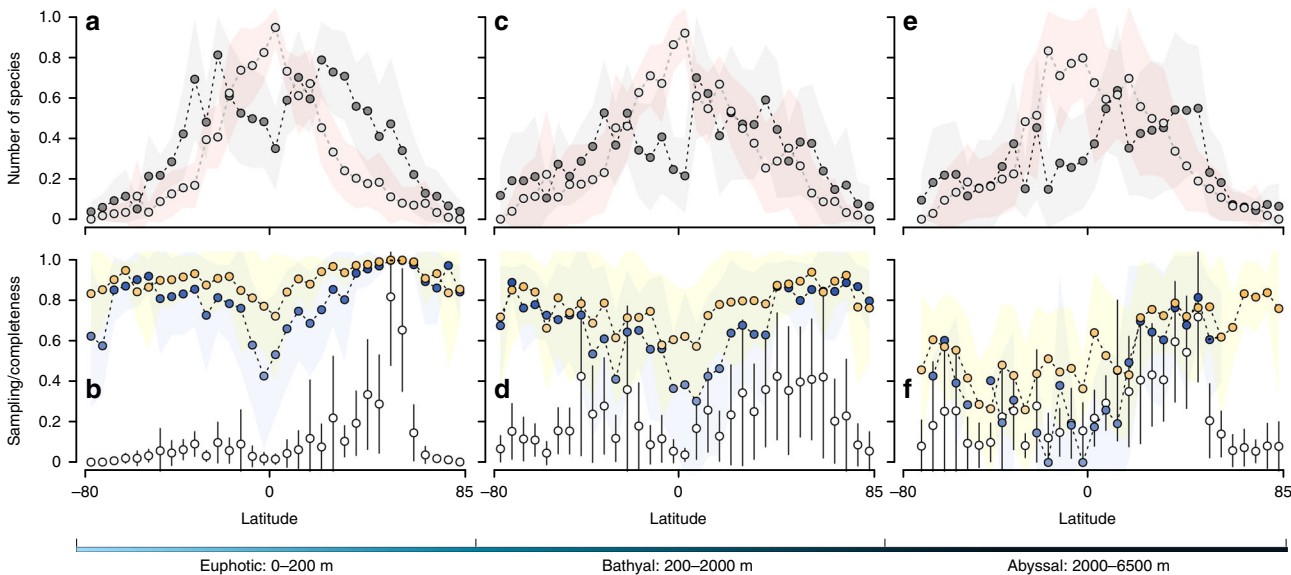

**Fig. 3** Patterns of latitudinal variation in species richness, spatial gaps and sampling efficiency at three different depth strata. Symbols and colors as in Fig. 2. **a**, **c**, **e** Mean (± s.d.) latitudinal variation in number of observed species and number of missing species at the euphotic, bathyal and abyssal strata. **b**, **d**, **f** Mean (± s.d.) latitudinal variation in sampling effort (number of unique sampling events) and two estimates of inventory completeness (sample coverage and species accumulation curve) at the euphotic, bathyal and abyssal strata. In all three depth strata the sampling effort decrease at the equator, reducing the level of completeness, number of observed species and increasing the number of spatial gaps

is hitherto no published account of this phenomenon as a general pattern at the species level. Indeed, if the antitropical distribution were a real biogeographical pattern, it would suggest that populations of many marine species were isolated in different hemispheres, on course for allopatric speciation. However, the accuracy of the antitropical pattern of species distribution requires adequate sampling effort at the tropics to confirm the true absence of missing species[21].

Many studies have already shown that sampling effort in the ocean is not evenly distributed across the globe[13,15,22,23]. Our results support such claims and show that there are indeed fewer sampling events near the equator, even for tropical taxa (e.g., stony corals; Supplementary Fig. 3). Although sampling events are also fewer at the poles, the smaller ocean area at higher latitudes[24] implies that the tropics are comparatively less sampled. Consequently, our estimates indicated a general decrease in inventory completeness toward the equator (similar to previous report for fishes[22]). The higher sampling intensity at mid-latitudes in both hemispheres is not surprising, given the higher funding for marine research provided by developed countries at these latitudes[22]. Conversely, most developing countries are tropical, with substantially less funding for marine biodiversity research, as indicated by the fewer number of research vessels and field stations[25]. Thus, it seems likely that increasing sampling effort in the tropics would increase estimated tropical marine diversity by discovering new species[25] and by filling the tropical gaps in the distribution of marine species, therefore invalidating the bimodal pattern and the tropical dip in species richness (see Supplementary Note 3; Supplementary Figs. 2, 17–18).

One of the hypotheses to explain antitropical distribution of marine species suggests that the high surface-water temperature and equatorial currents are effective barriers for species occurrence at the equator[26]. Therefore, this hypothesis predicts a decrease in spatial gaps (disjunct distribution) at higher ocean depths, where water is invariably colder[27]. However, our analyses of bathymetric strata revealed that spatial gaps in the tropics are not limited to shallow waters, but also occur in half of the deep-sea species. Thus, the apparent discontinuous distribution in

abyssal species may also be the result of insufficient knowledge of the deep-sea diversity[13,28]. In fact, our results revealed a substantial decrease in sampling events and estimates of inventory completeness at deep waters, probably reflecting the challenges of sampling those habitats[22]. We believe that low sampling effort in tropical deep waters contributes to the artefactual absence of species in two ways: first, by reducing the chance of recording the occurrence of tropical deep-sea species at low latitudes; second, by reducing the chance of recording shallow-water polar or temperate species that could submerge to deeper waters in tropical latitudes to escape from warm temperatures[18]. In addition, our results revealed that poor sampling effort in deep waters impacts how we perceive the latitudinal gradient of marine diversity, because the observed latitudinal pattern of marine species richness is a consequence of historically higher sampling effort at shallower ocean environments.

The latitudinal pattern of spatial gaps described here is based on the most complete data repository on the distribution of marine species[29], which has been increasingly used in analyses of marine biodiversity[11,16]. Preliminary analyses showed that adding extra data from nine other datasets on ophiuroids did not alter the general patterns described here (Supplementary Note 4; Supplementary Fig. 19; Supplementary Tables 1–2). Therefore, our results show a very consistent pattern of missing species occurrence associated with low inventory completeness at the tropics. The only groups with relatively high inventory completeness and spatial gaps across the tropics are the large vertebrate fishes (i.e., elasmobranchs and tuna fishes). Although the high completeness may indicate a real absence of fish species in the tropics, it is noteworthy that sampling effort at the equator was 30–40 times lower than at northern mid-latitudes for these groups (Supplementary Fig. 3). Interpolations showed that changes in sampling effort can alter considerably the tropical species richness for elasmobranchs and tuna fishes, despite the small difference in completeness estimate (Supplementary Figs. 2, 17). In addition, the number of absent species is highly reduced when using information from range maps based on expert knowledge and species distribution models

(Supplementary Fig. 1). Thus, the presence of spatial gaps in the tropics, associated with bimodal distribution of sampling events, may still suggest inefficiency of sampling effort for these groups. Finally, it is important to emphasise that our estimates of inventory completeness were conducted on a large spatial scale (latitudinal bands). Previous studies have shown that completeness tends to be high for large spatial scales, but decrease at finer spatial resolution[22,30] (see also Supplementary Fig. 6). Therefore, our knowledge about marine biodiversity at tropical latitudes is probably worse than reported here[22].

In conclusion, our results revealed that the dip in marine species diversity at the tropics is associated with a latitudinal gap in species distribution. This gap seems a consequence of low sampling at the tropics, although only the increase of sampling effort at low latitudes will confirm if the gradient in species absences is an artefact or a natural phenomenon. Improving the knowledge of marine biodiversity would require an international collaboration among research institutions of developed and developing countries[25], by sharing research facilities for field work (e.g., Brazil–Japan[31]), as well as by hosting specialists in taxonomic groups traditionally less studied (e.g., meiofauna[32]). As in Schrödinger's cat paradox, in which a hypothetical cat is simultaneously dead and alive, many marine species will remain both present (artefactual absence) and absent (real absence) in the tropics until the latitudinal bias in sampling effort is reduced. In the ecological version of the famous paradox, we will never know the real distribution of these species until we open and look inside the box of tropical oceans.

## Methods

**Species-occurrence data and quality control procedures**. We examined 10,072,204 records of Amphipoda, Bivalvia, Copepoda, Elasmobranchii, Gastropoda, Porifera, Rhodophyta, Scleractinia and Scombridae, obtained from the Ocean Biogeographic Information System (OBIS, http://www.iobis.org) and of Ophiuroidea obtained from the OBIS and additional nine datasets (Atlas of Living Australia, https://www.ala.org.au; Global Biodiversity Information Facility, https://www.gbif.org; Instituto de Ciencias del Mar y Limnología—Mexico, http://www.icmyl.unam.mx; Instituto de Investigaciones Marinas y Costeras—Colombia, http://www.invemar.org.co; MNHN Paris, http://www.mnhn.fr; NHM London, http://data.nhm.ac.uk; O'Hara et al.[33]; Scripps Institution of Oceanography, https://scripps.ucsd.edu; Smithsonian NMNH, https://naturalhistory.si.edu). We compiled additional data for ophiuroids to evaluate the representativeness of OBIS dataset in our further analyses. Data from OBIS represented 57.81% of all valid records for ophiuroids, and preliminary analysis showed that data from OBIS is consistent with other sources ($r > 0.89$; Supplementary Note 4; Supplementary Fig. 19; Supplementary Tables 1–2). Species occurrence records were obtained using the robis[34] and rgbif[35] packages implemented in the R software[36] or accessing directly the portals cited above. After retrieving the data, we eliminated records without information on the geographic coordinates, coordinates equal to zero, or records located inside continents (taken into account a 20 km buffer from the coastline). Valid coordinates were rounded to two decimal places to avoid inconsistency in numerical precision among datasets, as well as to avoid double counting sampling events. We selected only records at level of species, and eliminated records representing higher taxonomic levels. For records classified as subspecies we considered only the species name classification. We assessed the taxonomic validity of more than 36,500 species names by checking all the names at the World Register of Marine Species (WoRMS (http://www.marinespecies.org)). Whenever possible, misspelling or unaccepted species names were corrected. Records with species name not available or invalid (e.g., nomen dubium, inquirendum, temporary or nudum) were excluded. Records of fossil or non-marine species were also eliminated. Finally, we checked and excluded duplicate records (specimens with exact combination of scientific name, coordinates reference and sampling date) and records at latitudes lower than −80° (Antarctic continent) or higher than 85° (Arctic ice cap). Records with sampling date unavailable were kept if the scientific name and coordinates reference represented a unique combination in the dataset. Otherwise, the record was considered duplicate and excluded. After a thorough evaluation of data quality, a total of 3,442,702 valid records, of 34,849 species, were used in our study (see details of excluded data in Supplementary Note 5; Supplementary Fig. 20; Supplementary Tables 3–4; see Supplementary Fig. 21 for spatial distribution of sampling events).

**Sampling intensity and inventory completeness**. To quantify the latitudinal variation in sampling effort we calculated the total number of sampling events in each latitudinal band. A sampling event was defined as a unique combination of the location where a specimen was collected (latitude and longitude) and the sampling date. Additionally, we evaluated the sampling efficiency inside each latitudinal band by using two alternative methods to estimate inventory completeness: (1) we calculated the sample coverage, which is a measure of inventory completeness based on the total number of specimens recorded, and the number of rare species (singletons and doubletons)[5]. The high number of singletons and doubletons within a latitudinal band is probably an anomaly caused by undersampling[37] and indicates that many other species with low abundance or restricted spatial distribution (i.e., reduced probability of detection) may exist but have not yet been recorded. (2) We used the species accumulation curves (SACs) to build a measure of inventory completeness[16,38]. We measured the degree of curvilinearity of the last 10% of SACs for each latitudinal band to evaluate if the slope was shallow (saturation in the sampling) or steep (low completeness), as higher sampling effort in an area leads to a higher degree of curvature of its SACs. Because only the last 10% of SACs is used in the analysis, we applied this method only for latitudinal bands with more than 40 sampling events. Both methods follow the modifications of Stropp et al.[39] (see details in Supplementary Note 4; Supplementary Fig. 22), estimating inventory completeness between one (complete inventory) and zero (virtually inexistent inventory). Because inventory completeness may be overestimated at large spatial resolutions[22,30], as that used here, and our goal was to compare the sampling efficiency among latitudes, we only evaluate the latitudinal variation in inventory completeness, instead of using thresholds to categorise each latitudinal band as well or poorly sampled.

**Bathymetric strata**. To evaluate if the latitudinal variation in species richness, spatial gaps and sampling effort are consistent across marine depth strata we partitioned all the species records in three bathymetric strata: euphotic (0–200 m), bathyal (200–2000) and abyssal (2000–6500). These depth strata were selected to replicate the methodology used by Woolley et al.[12]. However, we used these alternative denominations, instead of continental shelf, upper slope and abyssal plane, to encompass pelagic species, for which sample depth may not necessarily reflect the depth of the sea floor. Depth information was extracted from all records for which sample depth was recorded. When sample depth was not available for a benthic species, we used the latitude and longitude of the record to determine the depth of the sea floor at that location using ETOPO1 1 arc-minute global bathymetry[40]. Overall, sample depth information was initially available for only 66.6% of the retrieved data. Using this additional data processing we were able to gather sample depth information for all benthic species and 90.08% of all records.

**Code availability**. The code to reproduce the observed and missing species richness analysis, as well as the inventory completeness and sampling effort estimates, is available at https://github.com/AndreMenegotto/SpatialGaps.

## Data availability

The primary data used here is freely available from OBIS (www.iobis.org) and additional data on Ophiuroidea distribution can be freely accessed from the cited sources. The processed data used in this study is available from the corresponding author upon request.

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

## Acknowledgements

A.M. is supported by a Ph.D. scholarship provided by Coordenação de Aperfeiçoamento de Pessoal de Nível Superior—Brasil (CAPES)—Finance Code 001. T.F.R. has been continuously supported by Conselho Nacional de Desenvolvimento Científico e Tecnológico (CNPq, grant PQ309550/2015-7). This project is also supported by INCT in Ecology, Evolution and Biodiversity Conservation, funded by MCTIC/CNPq (grant 465610/2014-5) and FAPEG (grant 201810267000023).

## Author contributions

A.M. and T.F.R. jointly conceived the ideas for the paper; A.M. compiled data, conducted analyses and wrote the first draft of the manuscript; A.M. and T.F.R. jointly revised the manuscript and responded to reviewers.

## Additional information

**Competing interests:** The authors declare no competing interests.

