## [Peer Review File · Nature Communications]

Reviewers' Comments:

Reviewer #1:

Remarks to the Author:

This MS by Menegotto and Rangel attempts to estimate latitudinal variation in sample completeness using a couple of different methods, and concludes that tropical biodiversity inventories are less complete than extra-tropical inventories. They infer from this that much of the apparent dip in richness in the tropics, which is apparent in databases of marine species occurrence, is an artefact of this less complete sampling in the tropics.

I am only familiar with a few studies that have addressed this apparent dip in richness, but it is true that those studies make little, if any, attempt to address this sampling problem. Thus, the present study, by drawing attention to, and making a first stab at estimating the magnitude of, this problem, has the potential to be an important contribution. There are a few issues, however, that I think should be addressed (or more thoroughly addressed), in this manuscript.

1. The authors explore a "Sousa-Baena" method, but then dismiss it because its estimates are not well correlated between the OBIS and "OBIS-PLUS" datasets that the authors explore. However, by itself, this is not a reason to conclude that S-B is inferior, much less so inferior that it should be omitted from the text almost entirely (except for Supp Fig 1). If the authors think this metric is flawed (or more flawed than the alternatives, like sample coverage), then they should say so, and justify this view. Otherwise, they should present all of their results, and allow the reader to form a judgment based on all the information.

2. I would like to know how robust the results are to the width of the "latitudinal bands" used to bin the data for analysis, since this is an arbitrary choice.

3. One potential explanation for more species absences in the tropical record could be greater biogeographic differentiation in the tropics. This raises a few potential problems. One is whether inventory completeness measures are robust to species having varying probabilities of occurrence across sample locations. If a larger proportion of species are geographically restricted, and some are widespread, this could create an uneven occurrence pattern of many species with zero, one, or very few occurrences but some who are widespread and still occur very broadly. Could this bias estimates of sample completeness lower in the tropics than in temperate regions. This issue requires some thought, and perhaps some investigation of the relative importance of differential unevenness in occurrence distributions (for instance, what if the temperate data are randomly subsampled to have the same number of samples as the tropics? Would this cause the difference in completeness to go away, consistent with the authors' hypothesis, or would the data still suggest undersampling in the tropics (indicating a differential evenness-driven trend in occurrence patterns)?

4. Is the difference in sample completeness of sufficient magnitude to explain the apparent dip in diversity? For instance, if the coverage estimates are used to extrapolate total richness (I think this is possible), what does the extrapolated richness pattern look like.

5. There is very strong hemispherical asymmetry (N vs S) in amount of sampling, and yet there's virtually no differences in estimated sample completeness. Can the authors explain this?

Reviewer #2:

Remarks to the Author:

This manuscript presents a spatial analysis of species presence-only data to draw conclusions about latitudinal gradients in marine biodiversity, arguing that a bimodal biodiversity gradient is an artefact of lower sampling effort at low latitudes rather than an ecological effect, and for the first time assessing marine latitudinal gradients in light of species absence. The manuscript is well put together, the analysis logical, and the data validation sensible, however I feel it could be significantly improved in two ways.

Firstly, the foundation of the work is based on the underlying assumption that species distribution is contiguous across space at a given scale (L66). This claim needs to be backed up either with citations to the literature or a solid and well-reasoned argument that can then form the foundation of the rest of the paper, and give weight to the idea that tropical absences are artefactual rather than true absences as a result of a tropical temperature barrier. One way of doing this may be to assess the distribution of the most common species, which are likely to be recorded even under low sampling effort, to see if they are contiguous, and if so this would provide a basis of an argument to assume all species ranges are contiguous.

Secondly, it would be very useful to be provided some information on the data that was not used, and some simple statistics on how the removal of this data effects the results – i.e. is number (or proportion per given latitude) of data-points removed significantly related to latitude, depth etc? This would ensure that it is data availability that is creating gaps, rather than the particular data cleaning method used, and could potential raise interesting questions about data quality in the tropics as well as availability.

Generally however, I feel that this manuscript is interesting and analysis is clear, and would be of significant interest to a general audience, as well as to ecologists and conservation scientists and practitioners.

Some further minor comments:

- L62: there are some techniques that do this - consider references to spatial applications of occupancy modelling, which can be used to identify where unobserved species are present, e.g.:
 - MacKenzie et al (2017). *Occupancy Estimation and Modeling: Inferring Patterns and Dynamics of Species Occurrence*, Elsevier
 - Dorazio et al (2006). *Estimating Species Richness and Accumulation by Modeling Species Occurrence and Detectability*, *Ecology* 87:842-854.
- L78: ref 10 does this with OBIS data, for example.
- L85: the bimodal pattern of marine biodiversity is the basis of this paper, however it is not explained as a concept in the introduction. Consider moving the first sentence of the discussion to the introduction.
- L93: the "(20, 25]" notation may be unfamiliar to some readers. Consider revising or explaining. Additionally, use notation consistently (e.g. L108, L110, L114-115)
- L200-201: argument requires citation.
- L210: consider rewording, to differentiate between oceans below 0-degrees latitude and the Southern Ocean.
- L215-216: to what extent is this a general behaviour, or Cnidaria specific?

- L217-218: I think this assertion requires a specific mention in the appropriate results section
- L276-277: removing coordinates equal to zero my result in removing records with latitudes rounded to the nearest whole degree – would therefore be useful to know how many of these there are.
- L304-305: the concept that more rare species results in increased likelihood of undiscovered rare species need further explanation.
- Fig 1, Supp. Figs 1, 2, 6-15: histograms include 3 grey shades, but only 2 are explained in the legends.
- Throughout (main text and supplementary material): occasional missing or un-needed words in sentences/unclear sentences/spelling mistakes. The manuscript and supplementary material would therefore benefit from a further thorough proof read.

Response to reviewers' comments

Reviewer #1 (Remarks to the Author):

This MS by Menegotto and Rangel attempts to estimate latitudinal variation in sample completeness using a couple of different methods, and concludes that tropical biodiversity inventories are less complete than extra-tropical inventories. They infer from this that much of the apparent dip in richness in the tropics, which is apparent in databases of marine species occurrence, is an artefact of this less complete sampling in the tropics.

I am only familiar with a few studies that have addressed this apparent dip in richness, but it is true that those studies make little, if any, attempt to address this sampling problem. Thus, the present study, by drawing attention to, and making a first stab at estimating the magnitude of, this problem, has the potential to be an important contribution. There are a few issues, however, that I think should be addressed (or more thoroughly addressed), in this manuscript.

- ✓ We thank the reviewer for his/her positive opinion about our manuscript and the constructive comments. We have strived to address all the reviewer's concerns and summarize them below.

1. The authors explore a "Sousa-Baena" method, but then dismiss it because its estimates are not well correlated between the OBIS and "OBIS-PLUS" datasets that the authors explore. However, by itself, this is not a reason to conclude that S-B is inferior, much less so inferior that it should be omitted from the text almost entirely (except for Supp Fig 1). If the authors think this metric is flawed (or more flawed than the alternatives, like sample coverage), then they should say so, and justify this view. Otherwise, they should present all of their results, and allow the reader to form a judgment based on all the information.

- ✓ We appreciate the suggestion. We now provide additional information and justify why the Sousa-Baena's method was not used in our study. We show that by analyzing the monotonic relationship between completeness estimate and number of records, this metric seems more susceptible to artefactual values of completeness than the other methods (see the new Supplementary Fig. 21). Although we believe that Sousa-Baena's method is really interesting and may be useful in other situations, it seems less appropriated for our data than the alternative completeness estimates. We have added a text explaining this result in

the end of the Supplementary Note 4. The monotonic relationship between each completeness estimate and the number of records is shown in the Supplementary Fig. 21.

2. I would like to know how robust the results are to the width of the “latitudinal bands” used to bin the data for analysis, since this is an arbitrary choice.

- ✓ We have replicated our analysis using two additional latitudinal bandwidths: 2° and 10°. The results of the replicated analyses are consistent with the 5° resolution that we used before. As speculated in the literature, we confirm that completeness estimate decrease at higher resolutions. We have added a couple sentences in the main text to describe this additional analysis, and the new results can be found in the Supplementary Fig. 6.

3. One potential explanation for more species absences in the tropical record could be greater biogeographic differentiation in the tropics. This raises a few potential problems. One is whether inventory completeness measures are robust to species having varying probabilities of occurrence across sample locations. If a larger proportion of species are geographically restricted, and some are widespread, this could create an uneven occurrence pattern of many species with zero, one, or very few occurrences but some who are widespread and still occur very broadly. Could this bias estimate of sample completeness lower in the tropics than in temperate regions? This issue requires some thought, and perhaps some investigation of the relative importance of differential unevenness in occurrence distributions (for instance, what if the temperate data are randomly subsampled to have the same number of samples as the tropics? Would this cause the difference in completeness to go away, consistent with the authors’ hypothesis, or would the data still suggest undersampling in the tropics (indicating a differential evenness-driven trend in occurrence patterns)?

- ✓ Species do have varying abundances over space, and therefore varying probability of occurrence in a random sample. However, we believe this is not a problem for completeness estimation, which does not assume that species must have equal abundance or an even spatial distribution. Such variation implies that more sampling effort is needed in some places than others, and places with similar sampling effort may have different completeness estimates. For example, if species are rarer in the tropics, many species should have reduced frequency and

probability of detection. Conversely, at latitudes with few widespread species the probability of recording common species must be higher. Thus, if sampling effort was homogenous across all latitudes, inventory of tropical species would continue being relatively less complete (Ecology 93(12): 2533-2547).

We think the idea of randomly reducing the sampling effort is a great opportunity to evaluate if species are rarer in the tropics than in high latitudes, and to explore the richness pattern in an equal sampling size scenario. We replicated the random subsampling 1000 times, calculating the average species richness and completeness estimate across all latitudes. Our results show that the reduction in sampling effort tends to decrease the completeness estimates. However, the estimated values for high latitudes still remain higher than those of the tropics (especially for sample coverage). This result suggests that while many tropical species are rare (few records), high latitudes species are equally abundant and well represented, keeping the completeness estimate relatively high, even with reduced sampling effort. Therefore, as expected, more samples are needed in the tropics than high latitudes to reach a high completeness estimation. We develop these ideas and explain the methods in a new supplementary note (Supplementary Note 3). The species richness and completeness estimate results were included in the Supplementary Fig. 2,4-5.

4. Is the difference in sample completeness of sufficient magnitude to explain the apparent dip in diversity? For instance, if the coverage estimates are used to extrapolate total richness (I think this is possible), what does the extrapolated richness pattern look like.

✓ This is a really good question!

To answer the reviewer we standardized the completeness estimate across all latitudes to compare the spatial variation in species richness. We also extrapolated the species richness to the maximum completeness estimate possible. As stated by Chao and Jost (Ecology 93(12): 2533-2547), such extrapolation should be extended only to a doubling of the reference sample size. Beyond that level (i.e., total completeness) the extrapolation would create unreliable estimations. The results show that when completeness is standardized the species richness tends to

be higher near the equator (Supplementary Fig. 17-18). Overall, spatial variation in species richness was similar between interpolated and extrapolated scenarios. Interestingly, when extrapolating, the species richness of some groups was higher at the tropical diversity dip than around the dip (see for example Ophiuroidea, Amphipoda, Porifera in Supplementary Fig. 17). We have included these results and all details about the analyses in the new Supplementary Note 3.

Thanks to this and the previous question we have realized that changes in sampling effort can alter considerably the tropical species richness for elasmobranchs and tuna fishes, despite relatively little difference in completeness estimate. For this reason, we have altered a brief passage in the discussion: ~~“Because sometimes units with low sampling effort may present artefactual high values of completeness~~ Interpolations showed that changes in sampling effort can alter considerably the tropical species richness for elasmobranchs and tuna fishes, despite the small difference in completeness estimate (Supplementary Fig. 2,17). In addition, the number of absent species is highly reduced when using information from range maps based on expert knowledge and species distribution models (Supplementary Fig. 1)”. Please, see the details in the main text.

5. There is very strong hemispherical asymmetry (N vs S) in amount of sampling, and yet there's virtually no differences in estimated sample completeness. Can the authors explain this?

- ✓ As mentioned in the answer of the question 4, similar sampling effort can produce different inventory completeness, just as different sampling effort can produce a similar inventory completeness. The similarity between northern and southern hemispheres, despite the strong asymmetry in number of sampling events, informs us that the sampling effort employed at each region are equally efficient to inventory the area, which may be associated with the low number of rare species. We have included the answer to this question in the new Supplementary Note 3.

Reviewer #2 (Remarks to the Author):

This manuscript presents a spatial analysis of species presence-only data to draw conclusions about latitudinal gradients in marine biodiversity, arguing that a bimodal biodiversity gradient is an artefact of lower sampling effort at low latitudes rather than an ecological effect, and for the first time assessing marine latitudinal gradients in light of species absence. The manuscript is well put together, the analysis logical, and the data validation sensible, however I feel it could be significantly improved in two ways.

- ✓ We are grateful to the reviewer for her/his positive comments about our manuscript and her/his very insightful suggestions.

Firstly, the foundation of the work is based on the underlying assumption that species distribution is contiguous across space at a given scale (L66). This claim needs to be backed up either with citations to the literature or a solid and well-reasoned argument that can then form the foundation of the rest of the paper, and give weight to the idea that tropical absences are artefactual rather than true absences as a result of a tropical temperature barrier. One way of doing this may be to assess the distribution of the most common species, which are likely to be recorded even under low sampling effort, to see if they are contiguous, and if so this would provide a basis of an argument to assume all species ranges are contiguous.

- ✓ We agree that range contiguity is a core assumption of our study, and we thank the reviewer for calling our attention to this important topic. Unfortunately, it is not possible to analyze the range contiguity based on species frequency, as suggested by the reviewer, because the frequency of records also seems spatially biased by irregular sampling effort. For example, among the five most frequent ophiuroid species, all with more than 5000 unique records, only one does not have gap. Such result could indicate that the most frequent species indeed do not have a contiguous latitudinal range. However, all these four species are over-recorded at well sampled latitudes. Specifically, more than 89% of their occurrence records came from only two latitudinal bands (between 50° and 60°). Even for the species with relatively less uneven distribution of sampling records, the frequency of records varies from a minimum of 2 at the equator (between 0° and 5°) to a maximum of 1535 at the well-sampled north (between 50° and 55°). For this reason, we believe that assessing the distribution of the most common species may not be a good method to evaluate the spatial contiguity of species range.

Instead, we explored the range contiguity of species with relatively well-known spatial distribution using range maps based on specialist knowledge (the IUCN range maps) and on ranges estimated through species distribution models. The results showed that, although some species apparently may have a disjunct spatial distribution, the proportion of species with spatial gaps in their latitudinal range is very low. Interestingly, the equatorial dip in species diversity tends to decrease, or even disappear, when using range maps. Therefore, this additional analysis suggests that the range contiguity assumption is not unrealistic. We have included the analysis and its results in the supplementary information (Supplementary Note 1; Supplementary Fig. 1).

Secondly, it would be very useful to be provided some information on the data that was not used, and some simple statistics on how the removal of this data effects the results – i.e. is number (or proportion per given latitude) of data-points removed significantly related to latitude, depth etc? This would ensure that it is data availability that is creating gaps, rather than the particular data cleaning method used, and could potential raise interesting questions about data quality in the tropics as well as availability.

- ✓ We now provide a table with the number of records that were kept, and the average proportion of records removed after each step of the data cleaning process (Supplementary Table 3). Overall, most of the data cleaning procedures had little impact on the initial dataset (< 2%), and will have minimal, if any, effect on the spatial gaps, mainly because some species are entirely removed from the dataset (e.g., fossil or freshwater species). The most impacting cleaning procedures was the exclusion of duplicates (33.81%) and records without identification to species level (26.51%). Although exclusion of duplicates will have no effect on the presence of gaps, we agree that the impact of the removal of records without species level identification would require further investigation. The predominance of such records in tropical waters could indicate, in fact, that tropical species absence may be caused by limitations on taxonomic expertise, instead of low sampling effort.

To evaluate the impact of removing records without species-level identification we repeated the retrieving of data from OBIS (only from OBIS for Ophiuroidea),

to explore the spatial distribution of unidentified records. This new dataset is not totally identical to our initial dataset (there was an increase of 7.1% in the number of records) but have a similar proportion of records without species identification (29.97%). The results showed that much of the unidentified records come from those well-sampled latitudes (Supplementary Fig. 20), and not the tropical latitudinal bands. While equatorial data (between -5° and 5°) represents only 2.49% of the records without identification, ten times more records are not identified at species level at those well-sampled latitudes (29.4% between 50° and 60°). Therefore, excluding records without identification to species level during the data cleaning process does not seem to be the cause of spatial gaps in the species range. In response to this comment we now discuss data cleaning statistics in the Supplementary Note 5.

Generally however, I feel that this manuscript is interesting and analysis is clear, and would be of significant interest to a general audience, as well as to ecologists and conservation scientists and practitioners.

✓ Thank you for the nice comment.

Some further minor comments:

- L62: there are some techniques that do this - consider references to spatial applications of occupancy modelling, which can be used to identify where unobserved species are present, e.g.:
 - MacKenzie et al (2017). *Occupancy Estimation and Modeling: Inferring Patterns and Dynamics of Species Occurrence*, Elsevier.
 - Dorazio et al (2006). *Estimating Species Richness and Accumulation by Modeling Species Occurrence and Detectability*, *Ecology* 87:842-854.

✓ Thanks for calling our attention to these important studies. We have rewritten the sentence in the main text to include both references as suggested by the reviewer.

- L78: ref 10 does this with OBIS data, for example.

✓ We respectfully disagree. In the L78 we stated that “few studies have explored the spatial distribution of species missing occurrence^{6,8} and, to our best knowledge,

no study has yet evaluated how the patterns of species absence may affect the richness estimate at global-scale”. The previously ref 10 (Plos One 5(8): e10223) does not do this. The paper explores the bathymetric distribution for the total number of OBIS records. The authors do not show any data or analysis of missing species distribution or species richness pattern.

- L85: the bimodal pattern of marine biodiversity is the basis of this paper, however it is not explained as a concept in the introduction. Consider moving the first sentence of the discussion to the introduction.

- ✓ We agree. We have now included an explanation of the bimodal pattern of marine biodiversity in the introduction.

- L93: the "(20, 25]" notation may be unfamiliar to some readers. Consider revising or explaining. Additionally, use notation consistently (e.g. L108, L110, L114-115).

- ✓ We removed the notation and standardized the description of the latitudinal bands.

- L200-201: argument requires citation.

- ✓ Thank you for the reminder. We have now added a reference.

- L210: consider rewording, to differentiate between oceans below 0-degrees latitude and the Southern Ocean.

- ✓ Thank you for this hint. We replaced “southern oceans” by “southern hemisphere”.

- L215-216: to what extent is this a general behaviour, or Cnidaria specific?

- ✓ Actually, that is a hypothetical situation for any marine taxa. The authors in the cited paper review the concept of bipolarity using studies with many marine groups (e.g., fishes). So, the conceptual model is not specific to Cnidaria. They only used Cnidaria and Radiolaria (Protozoa) data to investigate the phenomenon

(second part of the paper) because of their own expertise. All the details can be found in the original publication (Marine Biology Research 2(3): 200-241).

- L217-218: I think this assertion requires a specific mention in the appropriate results section.
 - ✓ Respectfully, we believe that the assertion is already mentioned in the results section. In the last paragraph of the results section we write: “We found that latitudinal patterns in species richness, spatial gaps, and sampling efficiency in the euphotic and bathyal strata were similar to that described for the entire ocean (Fig. 3a-d), as 89.23% of all records analysed came from shallow waters (Supplementary Note 2). However, at greater depths the spatial distribution of observed species richness was highly heterogeneous among the groups (Supplementary Figs 7-16), with no clear latitudinal pattern (Fig. 3e).”, supporting the assertion that: “our results revealed that poor sampling effort in deep waters impacts on how we perceive the latitudinal gradient of marine diversity, because the observed latitudinal pattern of marine species richness is a consequence of historically higher sampling effort at shallower ocean environments”. We hope that this point is clearer now.

- L276-277: removing coordinates equal to zero may result in removing records with latitudes rounded to the nearest whole degree – would therefore be useful to know how many of these there are.
 - ✓ Such records are probably from unknown positions, which have been auto-filled by zeros. However, we now show the number of records with 0-0 coordinates that were eliminated and how many species may have been affected by this data cleaning procedure. On average, the exclusion of records with 0-0 coordinates might affect less than 1% of the species with any spatial gap and, therefore, does not seem to have any effect in our results. We added a table describing this procedure in the supplementary material (Supplementary Table 4), and have included a discussion about this issue in the Supplementary Notes 5.

- L304-305: the concept that more rare species results in increased likelihood of undiscovered rare species need further explanation.

- ✓ We apologize for not being clear enough at this point. We now provide further explanation about the concept that more rare species results in increased likelihood of undiscovered rare species.

- Fig 1, Supp. Figs 1, 2, 6-15: histograms include 3 grey shades, but only 2 are explained in the legends.

- ✓ Thank you for reminding us. We now explain in the legend of all cited figures that “colors are mixed where histograms overlap”.

- Throughout (main text and supplementary material): occasional missing or un-needed words in sentences/unclear sentences/spelling mistakes. The manuscript and supplementary material would therefore benefit from a further thorough proof read.

- ✓ We have revised the entire manuscript and supplementary material. We hope we have eliminated all these problems in this new version.

Reviewers' Comments:

Reviewer #1:

Remarks to the Author:

Overall, I am satisfied with how the authors addressed my comments in the first round of review. I have three suggestions for additional changes.

1. the Discussion is a bit repetitive. It seems to me it could be shortened just by eliminating redundancy.
2. In my view Supplementary Figure 18 is the most interesting figure in the paper -- the estimated diversity gradient after standardizing sampling. This is much more interesting than the plots of observed and missing species, or sampling effort, which are currently in the main text. I would suggest moving this figure into the main text, even if it means moving one or more of the existing main text figures to the supplementary material. These figures should have confidence intervals placed on the estimates as well, so readers can have some sense of the potential estimation error associated with the gradient.
3. Moreover, I would very much like to also see the corresponding figures for the analyses where data were separated into different bathymetric categories. I am very interested to know what the diversity gradient looks like in shallow water versus deeper water, after standardizing for sampling (assuming that the confidence intervals aren't so wide that it is difficult to say much about the deeper water fauna.

Reviewer #2:

Remarks to the Author:

I feel that the authors have considered the concerns raised in my previous review in a thorough and logical way.

I would suggest an additional proof-read of the manuscript and supplementary information for some minor grammatical errors, however I feel that all of my previous comments have been satisfactorily addressed.

Response to reviewers' comments

Reviewer #1 (Remarks to the Author):

Overall, I am satisfied with how the authors addressed my comments in the first round of review. I have three suggestions for additional changes.

1. the Discussion is a bit repetitive. It seems to me it could be shortened just by eliminating redundancy.

- ✓ In fact, we detected redundancy in the beginning of the fifth paragraph. We have removed the first sentence and connected the remaining text with the previous (fourth) paragraph.

2. In my view Supplementary Figure 18 is the most interesting figure in the paper -- the estimated diversity gradient after standardizing sampling. This is much more interesting than the plots of observed and missing species, or sampling effort, which are currently in the main text. I would suggest moving this figure into the main text, even if it means moving one or more of the existing main text figures to the supplementary material. These figures should have confidence intervals placed on the estimates as well, so readers can have some sense of the potential estimation error associated with the gradient.

- ✓ The figure S18 is indeed very interesting. However, showing species richness under standardized sampling effort is not the main goal of the paper, as other studies have already used a similar technique in some degree. From the beginning of the study our main goal was to show that we can map the latitudinal absence of the species, and that such information can be used to evaluate how our ignorance is affecting the perception of macroecological patterns. For this reason, we prefer to keep the figure S18 as a supplemental information.
- ✓ We have now added confidence intervals in the interpolation plots of species richness (Fig. S2, S17) and inventory completeness estimates (Fig. S4-5).

3. Moreover, I would very much like to also see the corresponding figures for the analyses where data were separated into different bathymetric categories. I am very interested to know what the diversity gradient looks like in shallow water versus deeper water, after standardizing

for sampling (assuming that the confidence intervals aren't so wide that it is difficult to say much about the deeper water fauna.

- ✓ That would be fascinating! Unfortunately, the minimum sample coverage in tropical deep waters is too small to allow standardization of the sampling effort. As you can see below (Table 1) some groups would have to be interpolated to sample coverage values of 0.2, 0.1, or even lower. In addition, some groups with apparent high completeness (e.g., Scombridae) have poor species richness recorded in tropical deep waters and cannot be interpolated to even lower values. Therefore, we believe that interpolating the data of deep waters may produce unreliable and little informative results.

Table 1. Minimum sample coverage in tropical deep waters. < SC: minimum sample coverage for interpolation; < SCext: minimum sample coverage for extrapolation; Bands: number of latitudinal bands where the interpolation/extrapolation would be possible.

Taxon	< SC	< SCext	Bands
Ophiuroidea	0.393	0.488	32
Bivalvia	0.571	0.738	32
Gastropoda	0.236	0.342	32
Copepoda	0.292	0.448	32
Porifera	0.033	0.066	32
Rhodophyta	0.081	0.158	15
Amphipoda	0.010	0.134	32
Scleractinia	0.115	0.179	28
Elasmobranchii	0.667	0.772	19
Scombridae	1	1	14

Reviewer #2 (Remarks to the Author):

I feel that the authors have considered the concerns raised in my previous review in a thorough and logical way.

I would suggest an additional proof-read of the manuscript and supplementary information for some minor grammatical errors, however I feel that all of my previous comments have been satisfactorily addressed.

- ✓ We have revised again the entire manuscript and supplementary information.
We hope we have eliminated all these minor errors in this new version.